# Characterization of Biomarkers of Thrombo-Inflammation in Patients with First-Diagnosed Atrial Fibrillation

**DOI:** 10.3390/ijms25074109

**Published:** 2024-04-08

**Authors:** Julian Friebel, Max Wegner, Leon Blöbaum, Philipp-Alexander Schencke, Kai Jakobs, Marianna Puccini, Emily Ghanbari, Stella Lammel, Tharusan Thevathasan, Verena Moos, Marco Witkowski, Ulf Landmesser, Ursula Rauch-Kröhnert

**Affiliations:** 1Department of Cardiology, Angiology and Intensive Care Medicine, Deutsches Herzzentrum der Charité, 12203 Berlin, Germany; julian.friebel@dhzc-charite.de (J.F.); philipp.schencke@charite.de (P.-A.S.);; 2DZHK (German Centre for Cardiovascular Research), Partner Site Berlin, 10785 Berlin, Germany; 3Berlin Institute of Health at Charité—Universitätsmedizin Berlin, 10117 Berlin, Germany; 4Medical Department I, Gastroenterology, Infectious Diseases and Rheumatology, Charité—Universitätsmedizin Berlin, Corporate Member of Freie Universität Berlin and Humboldt-Universität zu Berlin, 12203 Berlin, Germany; 5Friede Springer Cardiovascular Prevention Center at Charité, Charité—Universitätsmedizin Berlin, Corporate Member of Freie Universität Berlin and Humboldt-Universität zu Berlin, 12203 Berlin, Germany

**Keywords:** atrial fibrillation, atrial myopathy, thrombo-inflammation, thrombin, protease-activated receptor 1, PAR1, tissue factor, MACE

## Abstract

Patients with first-diagnosed atrial fibrillation (FDAF) exhibit major adverse cardiovascular events (MACEs) during follow-up. Preclinical models have demonstrated that thrombo-inflammation mediates adverse cardiac remodeling and atherothrombotic events. We have hypothesized that thrombin activity (FIIa) links coagulation with inflammation and cardiac fibrosis/dysfunction. Surrogate markers of the thrombo-inflammatory response in plasma have not been characterized in FDAF. In this prospective longitudinal study, patients presenting with FDAF (*n* = 80), and 20 matched controls, were included. FIIa generation and activity in plasma were increased in the patients with early AF compared to the patients with chronic cardiovascular disease without AF (controls; *p* < 0.0001). This increase was accompanied by elevated biomarkers (ELISA) of platelet and endothelial activation in plasma. Pro-inflammatory peripheral immune cells (TNF-α^+^ or IL-6^+^) that expressed FIIa-activated protease-activated receptor 1 (PAR1) (flow cytometry) circulated more frequently in patients with FDAF compared to the controls (*p* < 0.0001). FIIa activity correlated with cardiac fibrosis (collagen turnover) and cardiac dysfunction (NT-pro ANP/NT-pro BNP) surrogate markers. FIIa activity in plasma was higher in patients with FDAF who experienced MACE. Signaling via FIIa might be a presumed link between the coagulation system (tissue factor-FXa/FIIa-PAR1 axis), inflammation, and pro-fibrotic pathways (thrombo-inflammation) in FDAF.

## 1. Introduction

Atrial fibrillation (AF) is the most common cardiac arrhythmia and has significant impacts on morbidity and mortality [1]. Recent trials have highlighted that patients with first- diagnosed AF (FDAF) (a very early phase of the condition) are considered to be at high risk of experiencing adverse outcomes [2]. The annual mortality rate after FDAF is 6.68% [3].

Thromboembolic risk (i.e., for ischemic stroke and peripheral embolism) is increased in patients with AF. Virchow’s triad (prothrombotic state, endothelial dysfunction, and abnormal left atrial blood flow) has been proposed as one explanation for left atrial appendage (LAA) thrombus formation and LAA-related thromboembolic events in patients with AF [4,5]. The pathological levels of hemostatic plasma markers in AF patients compared to those of healthy control subjects have led to the conclusion that these patients experience a generalized prothrombotic, hypercoagulable, and hypofibrinolytic state [6]. However, this has not been explicitly validated in patients with FDAF or in the absence of anticoagulation, since the data used to form this conclusion involve patients with advanced AF [7,8].

Over the last decades, studies and trials have focused on thromboembolic events and their prevention in AF patients; however, it has become evident that AF-related outcomes are significantly influenced by conditions beyond or preceding thromboembolic events [1]. Although the primary and secondary prevention of thromboembolic events has been sufficiently addressed by anticoagulation, the annual rate of major adverse cardiovascular events (MACEs) after FDAF is 10.78% [3]. Two aspects not addressed by anticoagulation require further attention. First, AF-related left ventricular dysfunction and heart failure (HF) are prevalent in 20–30% of patients with AF [1]. Rehospitalization/hospitalization due to decompensated HF occurs in 5.26% of patients after FDAF each year [3]. Second, the yearly prevalence of acute myocardial infarction (AMI) after FDAF is 0.71% [3]. The pathophysiology behind the unrelated thromboembolic adverse events in FDAF remains unclear.

The pathophysiological concept of thrombo-inflammation describes the complex interplay between hyper-coagulability/thrombogenicity, adverse cardiac remodeling, and inflammation [9,10]. Thrombo-inflammation plays a critical role in AF progression and MACE development [1,9,11,12,13].

Thrombin (FIIa) is the key effector protease of the coagulation cascade, mediating the activation of secondary hemostasis (FIIa drives fibrin deposition) and primary hemostasis (platelet activation) [10,14]. Protease-activated receptor-1 (PAR1) is the FIIa receptor on platelets, thereby coupling plasmatic and primary hemostasis. PARs comprise a family of four G protein-coupled receptors (PAR1–PAR4) [15,16].

Extravascular FIIa and PAR1 expressions have been observed in the endocardium, subendocardium, and myocardium of the left atria (LA) of AF patients [17]. On a cellular level, PAR1 is expressed in endothelial cells, vascular smooth muscle cells, cardiac fibroblasts, cardiomyocytes, and various immune cells [16]. Beyond its role in the coagulation cascade, FIIa facilitates non-hemostatic pro-fibrotic and pro-inflammatory effects via the proteolytic cleavage of PAR1. Signaling via the tissue factor (TF)-factor Xa (FXa)-FIIa-PAR1-axis has been proposed as a central mediator of thrombo-inflammation and is thus linked to the pathogenesis of atherosclerosis, cardiovascular inflammation, cardiac fibrosis, myocarditis, and heart failure (HF) [11,18,19,20,21,22,23]. However, these conclusions have been primarily derived from preclinical models.

Although it is known to occur during advanced AF stages, the thrombo-inflammatory response has not been characterized explicitly in FDAF. Therefore, we aimed to provide evidence of AF-associated thrombin activation, linking coagulation and platelet activation with downstream adverse phenotypes (i.e., cardiac fibrosis and inflammation) via PAR1 in patients presenting with early AF.

## 2. Results

### 2.1. Thrombin Generation and Activity in Patients with First-Diagnosed AF

Thrombin has been identified as a pivotal factor for thromboembolism in AF. However, its role in patients with FDAF—in whom secondary preventive efforts to modify the disease course are critical—remains poorly understood. Therefore, we compared 80 non-anticoagulated (because of very early sampling after hospital admission) consecutively enrolled patients without treatable causes of AF (e.g., hyperthyroidism, AMI, myocarditis, pericarditis, acute infectious disease, and acute inflammatory disease) who were admitted due to FDAF alongside 20 matched control patients that had comparable cardiovascular risk profiles (i.e., CHA_2_DS_2_-VASc, HF, BMI, diabetes) without AF.

We found that in the early stages of AF, FIIa generation was increased. This was indicated by the elevated plasma levels of prothrombin fragment F 1 + 2 (PTF) and thrombin–antithrombin complex (TAT) compared to the control patients (Figure 1A). FIIa activity was also increased in the FDAF cohort (Figure 1B). Likewise, the plasma levels of secondary hemostasis surrogates (fibrinogen and D-dimer) that are downstream of FIIa were higher in the patients with FDAF than in the control group (Figure 1B).

### 2.2. Plasma Biomarkers Suggest Increased Platelet Activation and Platelet–Endothelial Interactions during First-Diagnosed AF

Platelet activation has been reported in patients with advanced AF [6]. Anticoagulation focuses on the prevention of thromboembolic events [1]. However, activated platelets facilitate acute arterial thrombotic events (e.g., AMI) and release pro-fibrotic and pro-inflammatory mediators [24,25,26]. Therefore, we studied the biomarkers of primary hemostasis during FDAF.

In patients with FDAF, the biomarkers suggested a pro-thrombotic state (Figure 2). Platelet activation (through agonists such as FIIa) results in the release of P-selectin, chemokine ligand 4 (CXCL4)/platelet factor 4 (PF4), and CXCL7 (Figure 2A) [27]. P-selectin promotes platelet aggregation via platelet–fibrin and platelet–platelet binding [27]. Notably, biomarkers in plasma might result from the release of the effector molecules as well as the production of extracellular vesicles.

The initial steps of primary hemostasis (platelet adhesion) are facilitated by the von Willebrand factor (VWF), which is released from activated endothelium and megakaryocytes [27]. In patients with FDAF, we noted higher plasma values of VWF (Figure 2B). In these patients, increased VWF ristocetin cofactor activity corroborated the biological significance of the presumed platelet activation (Figure 2B). VWF is bound (and, therefore, inactive) to coagulation factor VIII (FVIII). FVIII is released from the complex with VWF by the action of FIIa. Here, we detected elevated plasma FVIII activity in the patients with FDAF (Figure 2C).

### 2.3. Thrombin Activity Is Associated with Platelet Activation Indicators in Patients with Early AF

The releases of FVIII and VWF from the complex by FIIa action suggest that FIIa coordinates secondary hemostasis and also initiates primary hemostasis by activating platelets in very early AF. This is supported by our finding that FIIa activity corresponds to surrogate markers of platelet activation, namely P-selectin, CXCL4/PF4, and CXCL7 (Figure 3).

### 2.4. Pro-Inflammatory Immune Cell Function in Very Early AF—Indications for PAR1 Activation via Thrombin

We have demonstrated that primary and secondary hemostasis (linked via FIIa) are activated during FDAF. Inflammatory immune processes are involved in the pathogenesis of AF and are associated with MACE (both thromboembolic and non-thromboembolic) [13,28]. Hence, we evaluated two prototypic indicators of pro-inflammatory immune responses in the context of FDAF. In our patient cohort with FDAF, the plasma levels of tumor necrosis factor α TNF-α and interleukin 6 (IL-6) were higher than in the control subjects (Figure 4A).

The next question we sought to answer was whether the pro-inflammatory response was linked to the activation of the thrombin receptor PAR1 on immune cells in patients with FDAF. FIIa-activated PAR1-positive peripheral blood mononuclear cells (PBMCs) (antibodies detected activated PAR1 cleaved by FIIa) were used to indicate augmented signaling through the TF-FXa/FIIa-PAR1 axis. In the patients with FDAF, a pro-inflammatory subset of immune cells, as indicated by the expression of TNF-α or IL-6, circulated more frequently than in the control patients (Figure 4B).

Further phenotyping revealed that different PBMC populations expressed PAR1. In patients with FDAF, antigen-presenting cells (HLA-DR^+^), non-T cells (CD3^−^), and lymphocytes expressing PAR1 were identified at higher levels than control subjects (Figure 4C–E). Furthermore, the surface distribution of PAR1 (=PAR1 that is accessible for activation via FIIa) on these cells was enhanced (Figure 4C–E). This indicates that in patients with FDAF, PAR1-positive immune cells circulate more frequently and exhibit higher receptor surface expression.

### 2.5. Thrombin Activity Correlates with Biomarkers of Cardiac Fibrosis and Cardiac Dysfunction in Patients with First-Diagnosed AF

Beyond mediating acute adverse vascular events, chronic low-grade inflammation promotes cardiac collagen deposition [11,19,28]. Atrial cardiomyopathy, also called atrial myopathy (AM), involves electrical, contractile/functional, and structural alterations that are associated with AF initiation and progression; therefore, AM can be considered to comprise substrate, triggers, and modifying factors [29]. Cardiac fibrosis is an important structural correlate of AM and HF [1]. We have previously reported higher plasma values of pro-fibrotic biomarkers in patients with FDAF [11,28]. Here, we have demonstrated that mediators of cardiac fibrosis, such as transforming growth factor beta (TGF-β), soluble interleukin 1 receptor-like 1 (sST2), and galectin 3 (Figure 5A), and surrogate markers of collagen synthesis, namely procollagen I C-terminal propeptide (PICP), procollagen I N-terminal propeptide (PINP), and procollagen III N-terminal propeptide (PIIINP) (Figure 5B), are positively correlated with the plasma activity of FIIa. In contrast, the C-telopeptide of type I collagen (ICTP), a biomarker of collagen degradation, was inversely associated with plasma FIIa activity (Figure 5C). FIIa activity was also positively associated with the plasma levels of natriuretic peptides, which are AM and HF surrogate markers (Figure 5D). This suggests a possible link between coagulation, cardiac fibrosis, and cardiac dysfunction in early AF.

### 2.6. Thrombin Activity Is Associated with Adverse Outcomes in Patients with First Diagnosis of AF

We have demonstrated that FIIa mediates thrombo-inflammation (i.e., the interplay of primary/secondary hemostasis, pro-inflammatory immune response, and pro-fibrotic remodeling) during early AF. Thrombo-inflammation has been linked to adverse outcome events in patients with cardiovascular disease [9,10]. Therefore, we tested the hypothesis that FIIa activity is specifically associated with the onset of MACE after FDAF.

Baseline plasma FIIa activity was higher in patients with FDAF who had experienced MACEs compared to patients without MACEs during follow-up (Figure 6A). The disease progression of AF and AM (indicated by unplanned re-hospitalization for AF and unplanned hospitalization for HF) corresponded to FIIa plasma activity (Figure 6B). Patients who had experienced thromboembolic events (transient ischemic attack [TIA] and ischemic stroke) and atherothrombotic complications (i.e., acute coronary syndrome [ACS]) after FDAF initially had higher levels of FIIa activity (Figure 6C).

## 3. Discussion

This biomarker-based study aimed to characterize the thrombo-inflammatory response during the very early stages of AF. By studying surrogate plasma markers in a cohort of patients presenting with FDAF, we noted several findings:Increased FIIa generation and FIIa activity in patients with FDAF.Platelet activation biomarkers correlate with FIIa activity.Pro-inflammatory immune cell function in very early AF is linked to PAR1 activation via FIIa.Plasma indicators of cardiac fibrosis and cardiac dysfunction are related to FIIa activity.FIIa activity is higher in patients with FDAF who experience adverse outcomes.

### 3.1. Plasma Biomarkers Indicate a Prothrombotic and Hypercoagulable State in Patients with First-Diagnosed AF

The current evidence, based on plasma biomarker analysis, suggests that a hypercoagulable state exists in patients with advanced AF [1]. This dysregulation of secondary hemostasis is indicated by elevated plasma levels of TF in patients with a history of AF > 6 weeks. Interestingly, one study found that TF levels remained high despite the use of vitamin K antagonist (VKA) for anticoagulation (with warfarin) [30]. Similarly, circulating active coagulation factors (TF and FXIa) have been detected despite oral anticoagulation with VKA [31]. These data highlight the persistent prothrombotic alterations predisposing individuals to adverse events [31].

Cardiovascular risk factors, cardiovascular comorbidities, and AF burden trigger, maintain, and accelerate hemostasis activation [32,33,34,35,36,37,38,39,40]. TF has been reported to be a central initiator and mediator of endothelial activation [36,37,38,39,40,41,42]. Endothelial TF overexpression has been found in the LA of patients with AF [43]. Furthermore, we have recently shown higher plasma TF levels to occur explicitly during FDAF [11].

AF (even in the absence of structural heart disease, hypertension, diabetes mellitus, coronary artery disease, or a history of stroke), cardiovascular risk factors, and cardiovascular disease (in the absence of AF) all result in the inflammatory activation of the endothelial layer [32,33]. This phenomenon can be observed locally (LA) and systemically. The activation of both primary and secondary hemostasis has been found to be more pronounced in the atria (especially the LAA) than in the periphery of patients with AF [4,32,44].

Endothelial activation and dysfunction are associated with increased TF expression, which subsequently leads to primary and secondary hemostasis activation. It has been proposed that both AF and its comorbid conditions increase thrombotic risk through endothelial dysfunction [32,43]. The physiological relevance of systemic endothelial dysfunction was demonstrated in AF patients by the reduction in reactive hyperemia measured via peripheral arterial tonometry [45].

In addition to coordinating secondary hemostasis, the key effector protease of the coagulation system, FIIa, drives the activation of human platelets through PAR1 [46]. AF episodes (especially tachycardic episodes) during FDAF might have an additional impact on both thrombogenicity and hypercoagulability, since patients observed during an early stage of AF (as expressed by classification as paroxysmal AF) when in sinus rhythm (SR), were shown to have P-selectin and TAT levels comparable to controls [32]. However, paroxysms were associated with early hemocoagulation changes [47]. Furthermore, the acute induction of AF during an electrophysiological study was found to increase P-selectin expression on platelets, platelet activation/function, and FIIa generation [48,49,50,51]. This is in line with a preclinical model, in which rapid atrial pacing downregulated the gene expression of thrombomodulin and of TF pathway inhibitor in the LA endocardium [52]. It has been suggested that AF-associated platelet aggregation and coagulation (FIIa generation) are positively associated with AF episode duration [53,54,55].

During more advanced stages of AF, increased FIIa generation and increased levels of a variety of primary and secondary markers of hemostasis activation that are not sufficiently counterbalanced by antithrombotic mechanisms have been observed compared to healthy controls [1].

Taken together, the existing evidence and our data indicate a clinical need for adequate therapeutic options that address FXa/FIIa upstream signaling to prevent thromboembolic and non-thromboembolic adverse events following FDAF.

### 3.2. Biomarkers of Inflammation and Cardiac Fibrosis in Early AF Are Linked to Thrombin

Next, we sought to understand how non-thromboembolic adverse events (e.g., AMI, HF, progression of AF) are linked to the activation of the hemostatic system. Cardiac fibrosis and cardiac inflammation are assumed to be important pathophysiological aspects of AF and are associated with AF-related morbidity and mortality [13,56]. Elevated plasma biomarkers of cardiac fibrosis have been demonstrated in patients with AF [56]. However, this concept has not been validated explicitly in the context of FDAF.

The degree of LA dysfunction and atrial myopathy in AF are correlated with a prothrombotic state [57,58]. Furthermore, the TF-FXa-FIIa-PAR1-axis has been linked to the pathogenesis of AF [59]. In preclinical models, FIIa participates in adverse LA remodeling and AM development [60]. Additionally, preclinical models have shown that the expression of key pro-fibrotic molecules, such as alpha-smooth muscle actin and TGF-β, and several pro-inflammatory effectors, such as IL-6, are coordinately induced by FIIa via PAR1 signaling pathways [19,61,62]. These models have shown that a hypercoagulable state can induce atrial fibrosis, further enhancing AF [19,60,62].

Here, we have provided evidence that FIIa is associated with pro-inflammatory and pro-fibrotic mediators in patients with FDAF. Although biomarkers of platelet activation have not been shown to be associated with thromboembolic events (e.g., cerebral infarction, TIA, or peripheral artery embolism) in AF, their role in AF-related outcome events (e.g., ACS and HF) and AF progression have not been studied [63]. Platelets release numerous pro-inflammatory and pro-fibrotic effectors [64]. As such, platelets are the major source of plasma TGF-β [65,66]. It has been shown that stimulation of platelet-rich plasma with thrombin led to increased TGF-β concentrations [19,67]. Here, we have demonstrated that plasma levels of TGF-β were associated with FIIa activity. Therefore, activation of primary hemostasis in patients with FDAF might be an important mediator of adverse events beyond thromboembolic complications.

### 3.3. Clinical Implications

The Atrial Fibrillation Better Care (ABC) approach in the 2020 AF guidelines from the European Society of Cardiology (ESC) comprises three goals: “A” anticoagulation/avoid stroke; “B” better symptom management; and “C” cardiovascular and comorbidity optimization [1]. One of the questions we sought to answer is how we could integrate this continuum into the concept of early thrombo-inflammation demonstrated in our study.

A recent prospective, multicenter observational study involving a 2-year follow-up (ANAFIE Registry) reported a relationship between coagulation biomarkers (D-dimer and thrombin generation) and clinical outcomes beyond classical thromboembolic events in elderly patients with AF [68]. It has been suggested that high coagulation pathway activity may contribute to these clinical events [68]. This highlights that thrombo-inflammation can explain the residual cardiovascular risk observed in patients with AF and its need to be therapeutically addressed.

Anticoagulation, which can be achieved via VKAs or non-VKA oral anticoagulants (NOACs), is the central therapeutic target for most patients with AF, in line with the “A” guideline of the ESC [1].

Endothelial dysfunction, platelet activation, and FIIa generation are present in patients with AF despite the use of VKAs [4,32]. A prospective randomized controlled trial demonstrated that the introduction of acetylsalicylic acid (ASA) plus low-dose warfarin (1 mg/d), low-dose warfarin alone (2 mg/d), or ASA plus low-dose warfarin (2 mg/d) did not reduce hemostatic markers, including VWF, fibrinogen, fibrin, and D-dimer [69]. Only full-dose VKA with warfarin (INR 2.0–3.0) reduced D-dimer and fibrinogen levels; however, this treatment did not reduce VWF or fibrin levels [70].

In contrast to NOAC treatment, coagulation factors synthesized during VKA treatment retain their capability to activate PARs. It is important to note that NOACs, not VKAs, are able to target PAR1 [11,19,59,60,71]. Likewise, VKA treatment, unlike NOAC treatment, has not demonstrated the ability to prevent adverse remodeling progression in preclinical trials of AF [60,62].

It has been suggested that FXa, which is upstream of FIIa, acts as a potent platelet agonist through PAR1 activation, driving arterial thrombosis [72]. The FXa antagonist rivaroxaban reduces platelet activation and thrombus formation [72]. FXa/IIa inhibition results in an antiplatelet effect that, together with its well-known potent anticoagulatory capacity, might lead to a reduced frequency of atherothrombotic events and improved outcomes in patients with AF [72]. The pleiotropic effects of FXa/FIIa inhibitors have been consistently demonstrated in various preclinical models [23]. Anticoagulants that target FXI/FXIa and FXII/FXIIa are currently under investigation for several thrombotic and non-thrombotic conditions [73,74]. However, their pleiotropic effects and capacity need to be evaluated in further trials before they are used clinically [75].

The initiation of anticoagulation in patients with device-detected atrial high-rate episodes (as possible precursor tachyarrhythmias) does not reduce the incidence of MACEs, as shown in NOAH-AFNET 6 [76]. The LOOP study tested whether early AF screening with implantable loop recorders and the use of anticoagulants prevented stroke in high-risk individuals. The authors concluded that an increase in AF detection and anticoagulation initiation was not associated with a significant reduction in the risk of stroke or systemic arterial embolism and that not all screen-detected AF merits anticoagulation treatment [77,78]. Regarding the implications for the timing of anticoagulation therapy, a meta-analysis reported that higher levels of coagulation factors (i.e., fibrinogen, PAI-1, and D-dimer) were associated with AF incidence in longitudinal studies, suggesting that a hypercoagulable state is associated with AF development [7]. In contrast, surrogate markers of endothelial activation (i.e., VWF) or platelet activation (i.e., P-selectin, β-thromboglobulin, and PF-4) were only associated with the prevalence of advanced AF [7]. This might reflect the different pathophysiological importances of primary and secondary hemostasis during various stages of AF. In line with these results, our data close a knowledge gap by providing insight into the earliest detectable phase of AF: FDAF. The clinical setting of FDAF refers to the moment when the AF burden exceeds a certain threshold, activating primary and secondary hemostasis to a degree that is sufficient to mediate adverse events.

A nationwide cohort study has revealed that early rhythm control therapy performed early after FDAF (<3 months) compared with usual care was associated with a lower risk of adverse events, including ischemic stroke, HF, AMI, and mortality, as well as composite endpoint [3]. From a pathophysiological perspective, restoring and maintaining SR and reducing AF burden are fundamental for targeting AF-associated thrombo-inflammation and preventing MACEs. Restoring SR and, therefore, physiological ventricular rates, has been shown to reduce biomarkers of endothelial activation and thrombin-related platelet activation [79,80,81,82]. In this regard, biomarkers of thrombo-inflammation, such as FIIa activity or PAR1 expression, might help to identify and monitor patients who are at higher risk for MACEs, in line with the ESC “B” goal.

Cardiovascular risk factors and comorbidities drive AF progression and trigger, maintain, and accelerate hemostasis activation and, consequently, thrombo-inflammation [32]. Therefore, it is crucial that AF risk factors are identified early and addressed to prevent the progression of AF substrate and thereby the onset of AF, in line with the ESC “C” goal.

### 3.4. Limitations

Due to the complexity of FDAF, our data cannot be generalized. We also deliberately refrained from designating a new pathophysiological model. Instead, we focused on a biomarker-based characterization of the thrombo-inflammatory phenotype in plasma during the earliest possible moment of blood sampling after the confirmation of AF to highlight that this population experiences a high risk of adverse events during follow-up. Platelet function, cardiac fibrosis, and cardiac dysfunction were studied in a biomarker-based analysis with inherent biases and limitations. To unveil the moment at which thrombo-inflammatory pathways are critically involved in the de novo progression of early AF and to assess the moment at which the hemostatic cascade needs to be addressed by anticoagulation warrants prospective trials that account for the different phenotypes (and risk factors) and include functional tests (e.g., thromboelastometry, flow cytometry, whole blood platelet function tests), cardiac magnetic resonance imaging, and endomyocardial biopsies.

## 4. Materials and Methods

### 4.1. Patient Studies

The local ethics committee (Charité—Universitätsmedizin Berlin) approved the study protocols, which were performed in accordance with the ethical principles of the Declaration of Helsinki. Each patient gave written informed consent before participating in this study. This prospective longitudinal study intended to explore the inflammatory pathomechanisms of AF.

Our cohort consisted of 100 consecutive patients. This cohort has been described previously [11,28]. Several inclusion criteria were applied to the AF group (*n* = 80): age ≥ 18 years; willing to sign a written informed consent form; admitted to our cardiology department due to a FDAF. Several exclusion criteria were applied to the AF group as well: a treatable/reversible cause of AF (e.g., hyperthyroidism, AMI, myocarditis, pericarditis, acute infectious disease, or acute inflammatory disease) or previous/ongoing anticoagulation treatment. Baseline peripheral blood samples and data were collected within the first 24 h of hospitalization (as soon as possible). Data were collected from unselected patients who were available for sampling, with no adjustment for confounding variables. Blood processing was performed immediately after sampling in non-anticoagulated patients [11,28].

During follow-ups, a MACE was defined as the occurrence of cardiovascular death, unplanned re-hospitalization for AF, or an unplanned hospitalization for HF, a TIA, an ischemic stroke, ACS, deep vein thrombosis, or peripheral thromboembolism [11,28].

The control group (*n* = 20) consisted of consecutive patients with comparable cardiovascular risk profiles without AF who had been admitted to our cardiology department. These patients were hospitalized due to hypertensive heart disease, HF, or elective coronary angiography [11,28].

A detailed description of the patient characteristics is provided in Table 1.

### 4.2. Laboratory Assays, Incl’ ELISA

Peripheral blood was collected in tubes (EDTA or citrate) (Vacutainer, BD Biosciences, Heidelberg, Germany) and stored in polypropylene tubes at −80 °C until use. Samples were centrifuged at 3000× *g* for 10 min or, in terms of biomarker of platelet activation, platelet-poor plasma was prepared. For PTF 1 + 2, TAT (both purchased from Abbexa Ltd., Cambridge, UK); thrombin activity (purchased from ASSAYPRO LLC, St. Charles, MO, USA); P-selectin, CXCL4, CXCL7, TGF-β, sST2, galectin 3, NT-pro ANP (all purchased from R&D Systems, Minneapolis, MN, USA); and PICP, PIIINP, ICTP, PINP (all purchased from Aviva Systems Biology, San Diego, CA, USA) ELISAs/assays were performed according to the manufacturer’s instructions. The intra-assay coefficient of variation (CV) was <10%, and inter-assay was CV < 10%. FIIa activity in plasma was measured with the Human Thrombin Chromogenic Activity Kit (ASSAYPRO LLC, St. Charles, MO, USA). The absorbance at 405 nm was read every 30 min for 2 h. Data were presented as change in absorbance per minute (∆A/min). Routine laboratory results (accredited Labor Berlin—Charité Vivantes GmbH, Department of Laboratory Medicine) were obtained from the patient’s medical records, and these included TNF-α (chemiluminescent immunoassay), IL-6 (electrochemiluminescence immunoassay), VWF–Ag, VWF–RCo (both latex-enhanced immunoassays), and FVIII (coagulometry).

### 4.3. Flow Cytometry

PBMCs were separated by Ficoll^®^ Paque in Leucosep tubes and washed twice in PBS containing 0.5% bovine serum albumin (BSA) (all from Sigma-Aldrich, St. Louis, MO, USA). Dimethyl sulfoxide cryo-conserved PBMCs were thawed in RPMI 1640 medium supplemented with PBS and fetal bovine serum (all from Sigma-Aldrich, St. Louis, MO, USA), washed, and resuspended in PBS/BSA. Cell suspensions were stained in PBS containing specific antibodies and 2% Beriglobin for 15 min at 4 °C, followed by washing in PBS and fixation in 4% paraformaldehyde for 10 min at 37 °C. Intracellular staining of cytokines was performed after fixation in the presence of 0.5% saponin (all from Sigma-Aldrich, St. Louis, MO, USA). The antibodies used were anti-human CD3 PE, IL-6 FITC (both from Becton, Dickinson and Company, Franklin Lakes, NJ, USA), HLA-DR Per-CP, PAR1-Alexa488 (both from R&D Systems, Minneapolis, MN, USA), and TNF-α APC Vio-770 (Miltenyi Biotec, Bergisch Gladbach, Germany). The percentage of FIIa-activated PAR1-positive (the antibody detects fragments of activated PAR1-cleaved-Ser42 protein) (Sigma-Aldrich, St. Louis, MO, USA) (visualized with goat anti-rabbit IgG Alexa 405; Thermo Fisher Scientific, Waltham, MA, USA) circulating PBMCs was measured by using flow cytometry with FACSCaliburTM and CellQuest software 5.1 (both from Becton, Dickinson and Company, Franklin Lakes, NJ, USA).

### 4.4. Statistical Analysis

Comparative (absolute) values of sST2, galectin 3, PICP, PINP, PIIINP, ICTP, and NT-pro ANP have previously been published [11,28]. Single comparisons were assessed by using the Mann–Whitney U test. Pearson’s coefficient was used for correlation analyses. All analyses were performed using GraphPad Prism version 9.3.0 software. The results are expressed as single values and medians. The overall α-level was 0.05. Nevertheless, we report a Bonferroni-adjusted α-level for the study in patients with FDAF (0.002) and for the FIIa correlation study in patients with FDAF (0.0042).

## 5. Conclusions

The prothrombotic state in FDAF indicates broader implications for its management. Intervening in this cascade might offer a synergistic approach to reducing disease progression and the vascular complications associated with AF. Targeting the TF-FXa-FIIa-PAR1-axis to reduce thrombo-inflammation might improve AF management strategies by combining proper anticoagulation (“A”); better symptom and rhythm control by reducing adverse remodeling AF progression and HF (“B”); and enabling the treatment of concomitant cardiovascular conditions and risk factors (“C”).

## Figures and Tables

**Figure 1 ijms-25-04109-f001:**
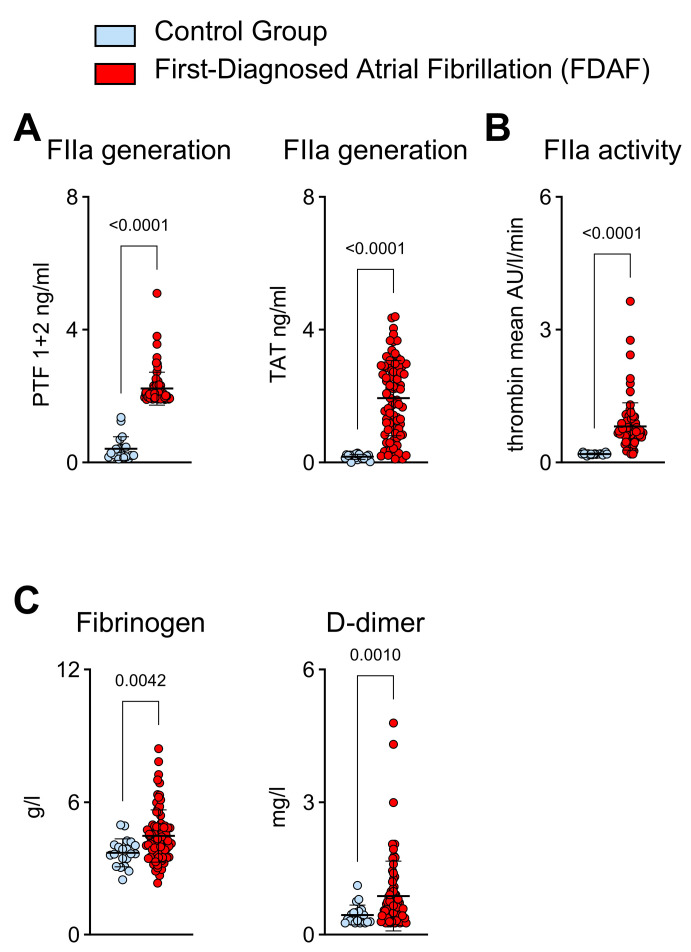
Thrombin generation and activity in patients with first-diagnosed atrial fibrillation (FDAF) showing increased plasma biomarkers of thrombin generation (prothrombin fragment F 1 + 2, PTF, and thrombin–antithrombin complex [TAT]) (**A**) and FIIa activity (**B**). Elevated corresponding surrogate plasma markers of the downstream activation of secondary hemostasis fibrinogen and D-dimer were noted (**C**). Patients with FDAF (*n* = 80) were compared to controls (patients with chronic cardiovascular diseases but without AF) (*n* = 20). Results are expressed as single values, median, and *p*-values.

**Figure 2 ijms-25-04109-f002:**
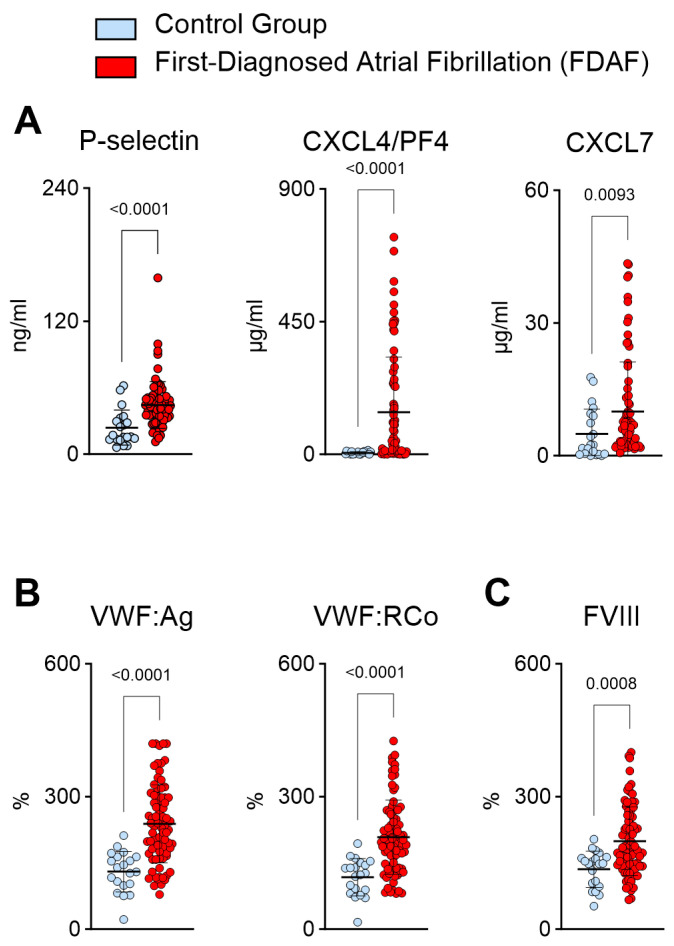
Plasma biomarkers suggest increased platelet activation during FDAF. Increased biomarkers of platelet activation (**A**) and platelet–endothelial interaction (**B**) were seen in FDAF patients. P-selectin, chemokine ligand 4, (CXCL4)/platelet factor 4 (PF4), CXCL7 (**A**); von Willebrand factor antigen (VWF–Ag), VWF ristocetin cofactor activity (VWF–RCo) (**B**). Elevated levels of the corresponding plasma surrogate marker of FIIa-related release of FVIII from VWF (as indicated by FVIIIa activity %) were also seen (**C**). Patients with FDAF (*n* = 80) were compared to controls (patients with chronic cardiovascular diseases but without AF) (*n* = 20). Results are expressed as single values, median, and *p*-values.

**Figure 3 ijms-25-04109-f003:**
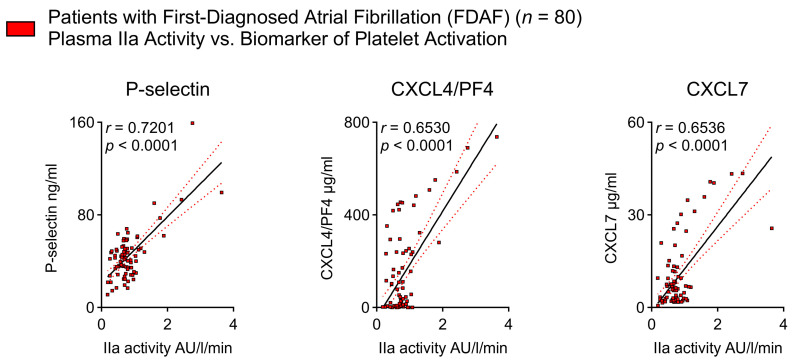
Thrombin activity is associated with indicators of platelet activation in patients with FDAF. This figure depicts plasma FIIa activity relative to plasma levels of surrogate markers of platelet activation (P-selectin, CXCL4/PF4, CXCL7). Results are expressed as single values (*n* = 80), Pearson correlation coefficients, and linear regression lines (black full lines) with 95% CI (red dotted lines) shown.

**Figure 4 ijms-25-04109-f004:**
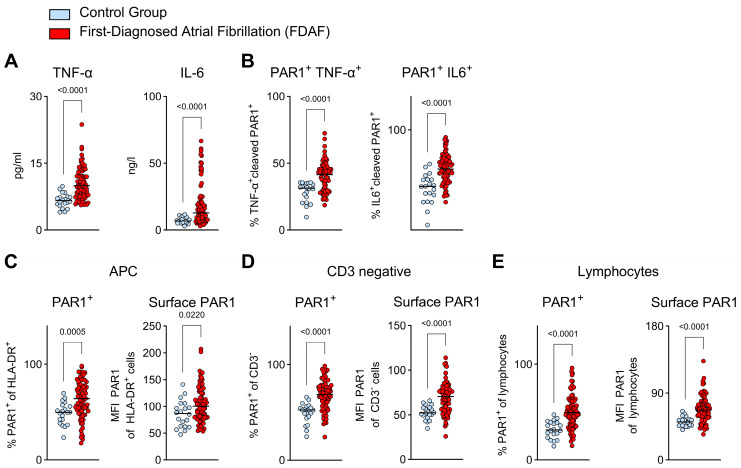
Pro-inflammatory immune cell function in FDAF—Indications for PAR1 activation via thrombin. (**A**) Increased plasma levels of pro-inflammatory mediators, namely tumor necrosis factor α (TNF-α) and interleukin 6 (IL-6), were seen in patients with FDAF. (**B**) Phenotyping via flow cytometry of peripheral blood mononuclear cells (PBMCs) revealed a higher percentage (% of total PBMCs) of circulating pro-inflammatory cells that possess thrombin-activated (cleaved) PAR1 (TNF-α^+^ or IL-6^+^ cleaved PAR1^+^) in patients with FDAF. The antibody detects fragments of activated PAR1-cleaved-Ser42 protein. Cytokines TNF-α and IL-6 were visualized via intracellular staining of permeabilized cells. (**C**–**E**) Evaluation of different immune cell subsets expressing PAR1 via flow cytometry. Antigen-presenting cells (APCs; HLA-DR^+^) (**C**), non-T cells (CD3^−^) (**D**), and lymphocytes (**E**). Left panel: percentage of a specified immune cell population expressing PAR1. Right panel: Increased mean fluorescence intensity (MFI) of PAR1. Extracellular (surface) PAR1 expression (=PAR1 that is accessible for activation via FIIa) was detected via staining of unpermeabilized PBMCs with a PAR1 antibody. Patients with FDAF (*n* = 80) were compared to controls (patients with chronic cardiovascular diseases but without AF) (*n* = 20). Results are expressed as single values, median, and *p*-values.

**Figure 5 ijms-25-04109-f005:**
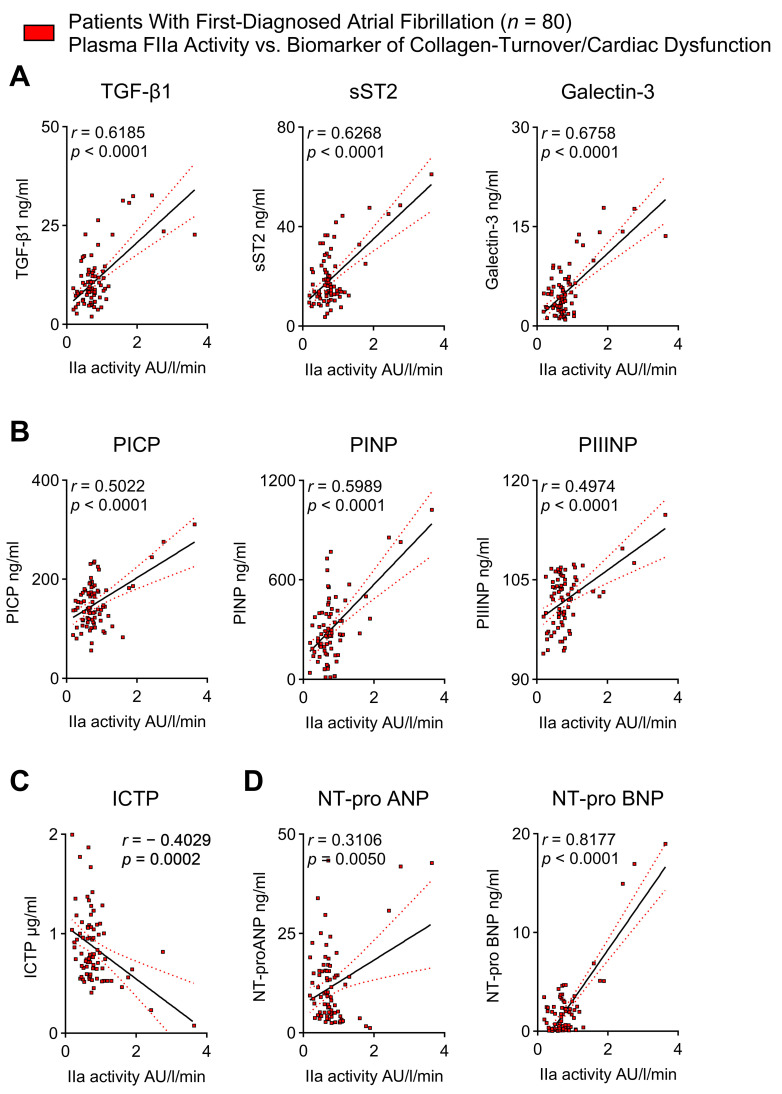
Thrombin activity in plasma correlates with biomarkers of cardiac fibrosis and cardiac dysfunction in patients with FDAF. Elevated FIIa activity corresponds with mediators of cardiac fibrosis (transforming growth factor beta [TGF-ß], soluble interleukin 1 receptor-like 1 [sST2], and galectin 3) (**A**), surrogate markers of collagen synthesis (procollagen I C-terminal propeptide [PICP], procollagen I N-terminal propeptide [PINP], and procollagen III N-terminal propeptide [PIIINP]) (**B**), biomarkers of collagen degradation (C-telopeptide of type I collagen [ICTP]) (**C**), and indicators of atrial myopathy and heart failure (N-terminal pro-atrial/brain natriuretic peptide [NT-pro ANP/NT-pro BNP]) (**D**) in the plasma of patients with FDAF. Comparative (absolute) values of sST2, galectin 3, PICP, PINP, PIIINP, ICTP, and NT-pro ANP have previously been published [11,28]. Here, we demonstrate for the first time an association of these biomarkers with FIIa generation. Results are expressed as single values (*n* = 80), Pearson correlation coefficients, and linear regression lines (black full lines) with 95% CI (red dotted lines) shown.

**Figure 6 ijms-25-04109-f006:**
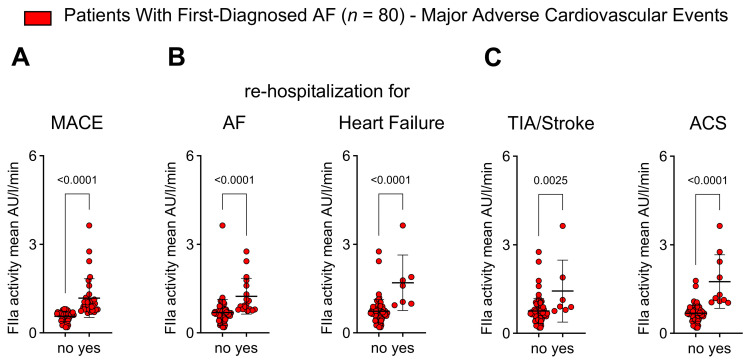
Plasma thrombin activity is associated with the onset of major adverse cardiovascular events after FDAF. Patients were stratified into two groups according to the occurrence of major adverse cardiovascular events (MACEs; no vs. yes) during follow-up after FDAF. Composite endpoint MACE: first occurrence of cardiovascular death, unplanned re-hospitalization for AF, unplanned hospitalization for heart failure, transient ischemic attack (TIA), ischemic stroke, acute coronary syndrome (ACS), deep vein thrombosis, peripheral thromboembolism (**A**). Adverse outcome events that are associated with AF progression (unplanned re-hospitalization for AF, unplanned hospitalization for HF) (**B**) and thromboembolic and atherothrombotic complications (TIA, ischemic stroke, ACS) (**C**) were related to FIIa activity at the time of FDAF. Results are expressed as single values (*n* = 80), median, and *p*-values.

**Table 1 ijms-25-04109-t001:** Baseline characteristics of patients with first-diagnosed AF and control patients.

	Control Group	Patients with FDAF	
	(*n* = 20)	(*n* = 80)	*p*-Value
Male/Female	50%/50%	62.5%/37.5%	n.s.
CHA_2_DS_2_-VASc	3.45	3.98	n.s.
History of Heart Failure	20%	26%	n.s.
Hypertension	85%	87.5%	n.s.
Age (years)			
<65	45%	27.5%	n.s.
65–75	30%	32.5%	n.s.
>75	25%	40%	n.s.
Diabetes	25%	30%	n.s.
History of TIA/Stroke	5%	10%	n.s.
Body Weight (kg)	82.95	85.93	n.s.
BMI kg/m^2^	27.97	27.55	n.s.
Previous or Ongoing Anticoagulation	0%	0%	n.s.
ASA	10%	10%	n.s.
FDAF			
Hospital discharge in SR	-	100%	
Follow-up			
Years	1	2.99	
MACE	0%	41.25%	
AF	0%	22.5%	
HF	0%	8.75%	
TIA/Stroke	0%	8.75%	
ACS	0%	12.5%	

Values are mean or %. Abbreviations: SR, sinus rhythm; n.s., not significant [11,28].

## Data Availability

Patient data is not publicly available due to general data protection regulations.

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
