# Peer review of "Characterization of Biomarkers of Thrombo-Inflammation in Patients with First-Diagnosed Atrial Fibrillation"

_ijms, 2024, doi:10.3390/ijms25074109_

Round 1

Reviewer 1 Report

Comments and Suggestions for Authors

This is a study on laboratory markers in a cohort of 100 consecutive with chronic cardiovascular disease with (FDAF; n = 80) or without AF (controls). The authors found (i) Increased in vivo FIIa generation and 'FIIa activity' in patients with FDAF.  (ii) Platelet activation biomarkers correlated with FIIa activity. 
(iii) Pro-inflammatory immune cell function  associated to PAR1 acti-vation via FIIa. (iv) Cardiac fibrosis and cardiac dysfunction are related to FIIa activity. (v) 'FIIa activity' is higher in patients with FDAF who experience adverse outcomes - Figures are fine.

Main issues:

Please specify what is new / REF 13 & 30? Such a sentence (L.445) is puzzling in this regard: Comparative values of sST2, galectin 3, PICP, PINP, PIIINP, ICTP, and NT-pro ANP have previously been published

Study design: Prospective? Substudy of REF 13 & 30? Whether it is or not prospective should be mentioned both in the text in the abstract. Cross-sectional or longitudinal?

Preanalytical aspects (see below)

What (circulating) FIIa activity is far from clear

L.411: Blood sampling was performed directly (should be ‘immediately’ or ASAP) after admission in non-anticoagulated patients: please specify when in the day (owing to circadian rhythms of the biomarkers).

Moreover it reads L. 408 that previous anticoagulation treatment was an exclusion criterion : previous or on-going ?

What about antiplatelet agents? (not mentioned in table 1)

Analysis: The investigators used a non-parametric test for significance but display the quantitative parameters as means ± SD

The authors should take into consideration the multiple testing for significance, and the number of investigated laboratory parameters / 80 patients

The authors should stick to the conclsuin that link between the coagulation system (tissue factor-FXa/FIIa-PAR1 axis), pro-inflammatory effector function, and pro-fibrotic pathways is only presumed.

Lab methods

Peripheral blood was collected in plasma [ ? shouldn’t it be into plastic?] tubes (Vacutainer, BD Biosciences); what was the anticoagulant?

We must know how plasma was prepared before freezing, and how plasma samples were thawed before testing.

Precision of the assays? Duplicates?

Regarding transmembrane proteins, do the immunoassays recognize both the genuinely soluble form (shedding) and the integral one in extracellular vesicles? e.g. P-selectin

Please expand on the ‘thrombin activity’ assay; description required; moreover it is not really plausible that there is circulating active thrombin in peripheral blood

Other points

Please clearly delineate which lab parameters were considered as (reliable) biomarkers of platelet activation in vivo. Note that it can't be vWF (main source being endothelium). The authors should exercise must caution about biomarkers of platelet activation, owing to the very high risk of surreptitious artefactual activation during sampling and plasma preparation.

Cardiac fibrosis and cardiac dysfunction: documented with imaging or based solely on biomarkers

The abstract is not informative enough :Nothing on lab methods; Results: no numerical results. The statement that Pro-inflammatory peripheral immune cell function was linked to activation via the FIIa receptor (protease-activated receptor 1, PAR1) is unsupported with authors’ original data. The authors can’t equal thrombo-inflammation with pro-fibrotic pathways.

Not mentioned in the abstract: tumor necrosis factor α (TNF-α) and interleukin 6 (IL-6); biomarkers of cardiac fibrosis and cardiac dysfunction; phenotyping via flow cytometry of peripheral blood mononuclear cells (PBMCs).

We do not need three references (4-6-8) of reviews from the same group the more recent one should suffice.

According to the two mentioned references 5 & 7, a systemic hypercoagulable state does not seem to exist.

Comments on the Quality of English Language

There are many odd phrasings such as

first diagnosis: should be first-diagnosed AF (FDAF) – OK in the title

FDAF, which correlates [ ?] to a very early phase of the condition

FDAF is associated with a high- L.17 risk patient group : high risk for what ?

‘pro-inflammatory effector function’

plasma levels of platelet activation (should be biomarkers of---)

4.2. ELISA/Assays : laboratory assays, incl’ ELISA

Author Response

We thank the reviewer for his time and effort assessing the previous version of the manuscript and providing useful comments to improve its value. We have addressed all the comments as explained below in the point-by-point reply. In accordance with the suggestion of reviewer #1 and #3, we adjusted the methodology section. We have also included a separate limitations section.

Comment 1:

Please specify what is new / REF 13 & 30? Such a sentence (L.445) is puzzling in this regard: Comparative values of sST2, galectin 3, PICP, PINP, PIIINP, ICTP, and NT-pro ANP have previously been published.

Reply:

We understand your concerns and have adjusted the text to be clearer. The previous publications have focused on absolute values, but not on a comparison with biomarker of FIIa activity which is the scoop of this manuscript.

Comment 2:

Study design: Prospective? Substudy of REF 13 & 30? Whether it is or not prospective should be mentioned both in the text in the abstract. Cross-sectional or longitudinal?

Reply:

Thank you very much for pointing this out. Our prospective longitudinal study intended to explore inflammatory pathomechanisms of AF. The previous publications focused on the aspects of cytotoxic T cells and intestinal barrier dysfunction. We have added this information as suggested by the reviewer.

Comment 3:

Preanalytical aspects: What (circulating) FIIa activity is far from clear.

Reply:

Thank you for catching this. We have adjusted the text to be clearer. Biomarker were analyzed in plasma samples from patients and controls.

Comment 4:

Preanalytical aspects: Blood sampling was performed directly (should be ‘immediately’ or ASAP) after admission in non-anticoagulated patients: please specify when in the day (owing to circadian rhythms of the biomarkers).

Reply:

The reviewer raises an interesting question. However, our intention was to generate data at the earliest possible clinical phase of FDAF. Therefore, baseline peripheral blood samples and data were collected within the first 24 h of hospitalization (= as soon as possible). Most of the patients were sampled during a very early phase of ED presentation. Blood processing was performed immediately after sampling.

Comment 5:

Preanalytical aspects: Moreover, it reads L. 408 that previous anticoagulation treatment was an exclusion criterion: previous or ongoing?

Reply:

Patients included in this study had to be not on previous or ongoing anticoagulation. We have added this information to the method section.

Comment 6:

Preanalytical aspects: What about antiplatelet agents? (not mentioned in table 1)?

Reply:

We thank the reviewer for this assistant comment and accordingly added ASA medication to the baseline characteristics. There was no significant difference in terms of patient characteristics between the two groups.

Comment 7:

Analysis: The investigators used a non-parametric test for significance but display the quantitative parameters as means ± SD.

Reply:

In accordance with the reviewer’s suggestion, results are now expressed as single values, median, and p-values.

Comment 8:

Analysis: The authors should take into consideration the multiple testing for significance, and the number of investigated laboratory parameters / 80 patients.

Reply:

We completely agree with the reviewer that small sample sizes always raise concerns about statistical power. We are aware of the pilot character of the data obtained from patients with FDAF. To address the reviewer’s concern, we added Bonferroni adjusted alpha-values in order to allow the readership an individualized interpretation of the data.

Comment 9:

Analysis: The authors should stick to the conclusion that link between the coagulation system (tissue factor-FXa/FIIa-PAR1 axis), pro-inflammatory effector function, and pro-fibrotic pathways is only presumed.

Reply:

Both you and reviewer #2 commented on this, so we are grateful to know that our current approach requires some rethinking.

Several statements that we made were more ambiguous than intended, and we have adjusted the text to be clearer. We would like to point out that it was not our intention to generalize our observation that biomarkers of thrombo-inflammation were increased in patients with FDAF.

We are aware that the pathophysiology of FDAF is much more complex and includes different phenotypes and various signaling pathways. We are also aware of the pilot character of the data obtained in human studies. To address the reviewer’s concern, we decided to add a new section with limitations at the end of the discussion.

Comment 10:

Lab methods: Peripheral blood was collected in plasma [shouldn’t it be into plastic?] tubes (Vacutainer, BD Biosciences); what was the anticoagulant?

Reply:

Thank you for catching this glaring and confusing error, which we have now corrected in the manuscript. We collected ETDA and citrated plasma. We run the assays with either EDTA or citrated plasma according to the manufacturer’s instructions.

Comment 11:

Lab methods: We must know how plasma was prepared before freezing, and how plasma samples were thawed before testing.

Reply:

Biomarker of platelet activation were measured in platelet-poor plasma. We have added this information to the method section.

Comment 12:

Lab methods: Precision of the assays? Duplicates?

Reply:

Intra-assay coefficient of variation (CV) was < 10%, inter-assay CV was < 10%. According to the available plasma volume, samples were run as single approaches, duplicates, or triplicates.  

Comment 13:

Lab methods: Regarding transmembrane proteins, do the immunoassays recognize both the genuinely soluble form (shedding) and the integral one in extracellular vesicles? e.g. P-selectin

Reply:

The reviewer raised an important question. We agree that additional analyses would provide useful and important data, but we believe that the recommended analyses are outside the scope of this study. We intended to characterize a biomarker profile in patients with FDAF that might (in the future) to help making a risk stratification in this specific group. We are aware of the pilot character of the data obtained from presumed indicators of platelet activation which we have now addressed in a new limitations section. As suggested by the reviewer, we have also highlighted that biomarkers in plasma might result from the release of the effector molecules as well as the production of extracellular vesicles.

Comment 14:

Lab methods: Please expand on the ‘thrombin activity’ assay; description required.

Reply:

Both you and reviewer #3 noted this concern. Therefore, we have adjusted the text to be clearer. FIIa activity in plasma was measured with the Human Thrombin Chromogenic Activity Kit (Assaypro LLC). The absorbance at 405 nm was read every 30 minutes for 2 hours. To generate a standard curve from the optimal reaction time, a graph was plotted using the standard concentrations on the x-axis and the change in absorbance per minute (∆A/min) on the y-axis after subtracting the background. The best fit line was determined by regression analysis of the 4-parameter curve. The unknown sample concentration was determined from the standard curve and the value was multiplied by the dilution factor. The conversion of WHO U and IU is 1 WHO U/ml = 1.2 IU/ml. The conversion of AU and IU is 1 AU/ml = 0.15 IU/ml.

Comment 15:

Moreover, it is not really plausible that there is circulating active thrombin in peripheral blood.

Reply:

Thank you for catching this glaring and confusing error, which we have now corrected in the manuscript.

Comment 16:

Please clearly delineate which lab parameters were considered as (reliable) biomarkers of platelet activation in vivo. Note that it can't be vWF (main source being endothelium). The authors should exercise must caution about biomarkers of platelet activation, owing to the very high risk of surreptitious artefactual activation during sampling and plasma preparation.

Reply:

Several statements that we made were more ambiguous than intended, and we have adjusted the text to be clearer.

Comment 17:

Cardiac fibrosis and cardiac dysfunction: documented with imaging or based solely on biomarkers?

Reply:

We agree that additional analyses would provide useful and important data. However, we believe that the recommended analyses are outside the scope of this study. Previous studies have shown a robust correlation of cardiac fibrosis in EMBs with circulating biomarkers of fibrosis.  We are aware of inherent limitations which we have now added to the discussion section.

Comment 18:

The abstract is not informative enough: Nothing on lab methods; Results: no numerical results. The statement that Pro-inflammatory peripheral immune cell function was linked to activation via the FIIa receptor (protease-activated receptor 1, PAR1) is unsupported with authors’ original data. The authors can’t equal thrombo-inflammation with pro-fibrotic pathways.

Reply:

We agree and have revised accordingly.

Comment 19:

Not mentioned in the abstract: tumor necrosis factor α (TNF-α) and interleukin 6 (IL-6); biomarkers of cardiac fibrosis and cardiac dysfunction; phenotyping via flow cytometry of peripheral blood mononuclear cells (PBMCs).

Reply:

We agree and have revised accordingly (with regard to the word limit of IJMS for the abstract that is 200).

Comment 20:

We do not need three references (4-6-8) of reviews from the same group the more recent one should suffice.

Reply:

We agree.

Comment 21:

According to the two mentioned references 5 & 7, a systemic hypercoagulable state does not seem to exist.

Reply:

Thank you very much for pointing this out. Motoki et al. and Breitenstein et al. focused only on AF patients (not on controls as well) and specifically on the LAA. Ref. 5 & 7 were included to show that the LA/LAA contributes significantly to the thromboembolic events in AF patients. The statement that pathological levels of hemostatic plasma markers in AF patients compared to those of healthy control subjects have led to the conclusion that these patients experience a generalized prothrombotic, hypercoagulable, and hypofibrinolytic state, relates to Ref. PMID: 32675218.

However, our main intention was to phenotype patients during a very specific phase of AF (FDAF). AF (even in the absence of structural heart disease, hypertension, diabetes mellitus, coronary artery disease, or a history of stroke), cardiovascular risk factors, and cardio-vascular disease (in the absence of AF) all result in the inflammatory activation of the endothelial layer. This phenomenon can be observed locally (LA) and systemically. The activation of both primary and secondary hemostasis has been found to be more pronounced in the atria (especially the LAA) than in the periphery of patients with AF (PMID: 37063762, PMID: 29759366). However, previous studies have focused on various stages of AF, but not explicitly on FDAF.

Comment 22:

Comments on the Quality of English Language: There are many odd phrasings such as

  • first diagnosis: should be first-diagnosed AF (FDAF) – OK in the title
  • FDAF, which correlates [?] to a very early phase of the condition
  • FDAF is associated with a high- L.17 risk patient group : high risk for what ?
  • ‘pro-inflammatory effector function’
  • plasma levels of platelet activation (should be biomarkers of---)
  • 2. ELISA/Assays : laboratory assays, incl’ ELISA

Reply:

We thank the reviewer in supporting our intention and have revised the manuscript accordingly.

Reviewer 2 Report

Comments and Suggestions for Authors

The manuscript entitled “Characterization of Thrombo-inflammation in Patients with First-Diagnosed Atrial Fibrillation” reported the role of FIIa activity in the First-Diagnosed Atrial Fibrillation. They have clarified the PAR1 signaling as a presumed link among the coagulation system (tissue factor-FXa/FIIa-PAR1 axis), pro-inflammatory effector function, and pro-fibrotic pathways (thrombo-inflammation) in FDAF. They assumed that this cascade might offer a synergistic approach to reducing disease progression and the vascular complications associated with AF. This study used patients blood samples to build this correlation. I have the following concerns.

1. It is still very difficult to make a decision that FIIa activity is important in the FDAF, since the authors only provided the high correlation between FIIa activity and FDAF rather than any artificial supplementation to animal models. I hope the authors can provided more evidences.    

2. A systematic diagram summarizing their conclusion should be provided at the end of the manuscript.

Author Response

We thank the reviewer for his time and effort assessing the previous version of the manuscript and providing useful comments to improve its value. We have addressed all the comments as explained below in the point-by-point reply. In accordance with the suggestion of reviewer #1 and #3, we adjusted the methodology section. We have also included a separate limitations section.

Comment 1:

It is still very difficult to make a decision that FIIa activity is important in the FDAF, since the authors only provided the high correlation between FIIa activity and FDAF rather than any artificial supplementation to animal models. I hope the authors can provided more evidences.  

Reply:

Several statements that we made were more ambiguous than intended, and we have adjusted the text to be clearer. Due to the complexity of FDAF our data cannot be generalized. We also deliberately refrained from designating a new pathophysiological model. Instead, we focused on a biomarker-based characterization of the thrombo-inflammatory phenotype in plasma during the earliest possible moment of blood sampling after the confirmation of AF and to highlight that this population experiences a high risk of adverse events during follow-up. Platelet function, cardiac fibrosis, and cardiac dysfunction were studied in a biomarker-based analysis with inherent biases and limitations To unveil the moment in which thrombo-inflammatory pathways are critically involved in the de-novo progression of early AF and to assess the moment where the hemostatic cascade needs to be addressed by anticoagulation, warrants prospective trials that account for the different phenotypes (and risk factors) and include functional tests (e.g. thromboelastometry, flow cytometry, whole blood platelet function tests), cardiac magnetic resonance imaging, and endomyocardial biopsies.

Previous studies (that have all used biomarkers) have focused on advanced stages of AF, but, thus far, surrogate markers of the thrombo-inflammatory response in plasma have not been characterized especially in patients with FDAF. These studies mostly included healthy controls and not patients with a comparable cardiovascular risk profile as done in our study. Furthermore, data obtained from (artificial) animal models have linked thrombo-inflammation with adverse cardiac remodeling and, to some extent, with de-novo AF (precursors). However, this assumption has not been studied thus far in patient cohorts with AF.

Comment 2:

A systematic diagram summarizing their conclusion should be provided at the end of the manuscript.

Reply:

We thank the reviewer in supporting our intention. In order with your suggestion, we included a systematic diagram to summarize our conclusions.

Reviewer 3 Report

Comments and Suggestions for Authors

The authors have published several papers describing this cohort before. The present paper presents data on activation of thrombin, PAR1 and platelets, connected to inflammation and fibrosis in patients with first diagnosis of atrial fibrillation.

The study shows that platelet activation markers and thrombin generation and activity is associated with FDAF. Thrombin activity in the FDAF patients is shown to correspond to platelet markers, and markers of fibrosis and HF.  Further that PBMCs (and subpopulations) have higher expression of both PAR1 and activated PAR1 in FDAF compared to the control group. Also inflammatory markers (in plasma or intracellular) are associated with the FDAF-group.

The impression is that some manuscript preparation has been done by copy-paste and only superficial proofreading of the thirteen co-authors. Unfortunately resulting in some flaws, e.g. with totally misleading reference referral. Except from that, the study seems to be solid, but the manuscript needs to be corrected.

(L82): The abbreviation HF is used in introduction without explanation. In figure 6 “heart failure” is used as panel text even though HF abbreviation is explained in figure legend and abbreviations has been used for other follow up events. Please correct.

L313 (Discussion): misspelling AFDF instead of FDAF?

L317-8 (Discussion): Reformulate sentence, Ref66 (which is a review) has not measured FIIa.

Methods:

*specify how plasma was prepared

*correct the description of flow cytometry, all reference numbering in this section is wrong since the text is a copy-paste from an earlier publication (probably from Friebel et al. Cells, 2022).

Even if correcting for the reference as pointed out above, the reference for intracellular staining is not directly pointing to a paper describing the method. Please specify the procedure in the present manuscript or include reference actually using intracellular staining.

*how was the activated PAR1 detected by flow cytometry? The antibody from Sigma is unconjugated. Was it stained with a secondary antibody or conjugated? Include more details.

*include the method for measurement of TNF and IL6 in plasma, this is hardly from medical records

*include also the methods for vWF:Ag, vWF:RCo and FVIII, same reason as for TNF and IL6 measurements

*explain how the thrombin activation was calculated. Is it really expressed as AU/L/minute (figure 1, 3, 5 and 6)? From the protocol supplied be the producer one would expect AU/L (or mL).

Figure 2: units for CXCL4(PF4) and CXCL7 should be checked, probably it should be ng/mL not µg/mL. The presented plasma level of PF4 seems to be 15 times higher than what is expected in serum (and 1500 times higher than normal plasma).

Figure 4: both panel titles and figure legend need to be corrected. Quite confusing  as presented. Left and right panel of C,  D and E each represent APCs, CD3- and lymphocytes, respectively. But at first glance it looks like this hold true for the left part of each panel. More other the terms “PAR1+” and “Activated PAR1” are both used in the figure but when reading the figure legend one gets the impression that both terms refer to PAR1 that has been cleaved. In the methods section a general PAR1-antibody is listed but has it been used? Is there actually any results or figure including flow cytometric results on PAR1-positiv cells (including cells positive for PAR1 independent of activation)? The figure legend needs some improvement – the “(A)” and “(B)” etc have not been placed correctly.

Table 1: include all events in the follow-up period, now only MACE has been listed. It should correspond to figure 6.

Author Response

We thank the reviewer for his time and effort assessing the previous version of the manuscript and providing useful comments to improve its value. We have addressed all the comments as explained below in the point-by-point reply. In accordance with the suggestion of reviewer #1 and #3, we adjusted the methodology section. We have also included a separate limitations section.

Comment 1:

The impression is that some manuscript preparation has been done by copy-paste and only superficial proofreading of the thirteen co-authors. Unfortunately resulting in some flaws, e.g. with totally misleading reference referral. Except from that, the study seems to be solid, but the manuscript needs to be corrected.

Reply:

We thank the reviewer in supporting our intention. The manuscript has been restructured and rewritten in order to address this important concern and to clarify our main conceptual axis.

Comment 2:

(L82): The abbreviation HF is used in introduction without explanation. In figure 6 “heart failure” is used as panel text even though HF abbreviation is explained in figure legend and abbreviations has been used for other follow up events. Please correct.

Reply:

We thank the reviewer for this assistant comment and have revised the text accordingly.

Comment 3:

L313 (Discussion): misspelling AFDF instead of FDAF?

Reply:

Thank you. It has been corrected.

Comment 4:

L317-8 (Discussion): Reformulate sentence, Ref66 (which is a review) has not measured FIIa.

Reply:

We agree. In order with your suggestion, we have adjusted the text to be clearer.

Comment 5:

Methods: specify how plasma was prepared.

Reply:

The reviewer raises an important question. However, our intention was to generate data at the earliest possible clinical phase of FDAF. Therefore, baseline peripheral blood samples and data were collected within the first 24 h of hospitalization (= as soon as possible). Most of the patients were sampled during a very early phase of ED presentation. Blood processing was performed immediately after sampling. We collected ETDA and citrated plasma. We run the assays with either EDTA or citrated plasma according to the manufacturer’s instructions. Biomarker of platelet activation were measured in platelet-poor plasma. We have added this information to the method section.

Comment 6:

Methods: correct the description of flow cytometry, all reference numbering in this section is wrong since the text is a copy-paste from an earlier publication (probably from Friebel et al. Cells, 2022).

Reply:

Thank you for catching this glaring and confusing error, which we have now corrected in the manuscript.

Comment 7:

Methods: Even if correcting for the reference as pointed out above, the reference for intracellular staining is not directly pointing to a paper describing the method. Please specify the procedure in the present manuscript or include reference actually using intracellular staining.

Reply:

We totally agree. In accordance with the reviewer’s suggestion, we have extended the description in the methods section. PBMC were separated by Ficoll® Paque in Leucosep tubes and washed twice in PBS containing 0.5% bovine serum albumin (BSA) (all from Sigma-Aldrich, St. Louis, MO, USA). Dimethyl sulfoxide cryoconserved PBMCs were thawed in RPMI 1640 medium supplemented with PBS and fetal bovine serum (all from Sigma-Aldrich, St. Louis, MO, USA), washed and resuspended in PBS/BSA. Cell suspensions were stained in PBS containing specific antibodies and 2% Beriglobin for 15 min at 4°C, followed by washing in PBS, and fixation in 4% paraformaldehyde for 10 min at 37°C. Intracellular staining of cytokines was performed after fixation in the presence of 0.5% saponin (all from Sigma-Aldrich, St. Louis, MO, USA). Antibodies used were anti-human CD3 PE, IL-6 FITC (both from Becton, Dickinson and Company, Franklin Lakes, NJ, USA), HLA-DR Per-CP, PAR1-Alexa488 (both from R&D Systems, Minneapolis, MN, USA), and TNF-α APC Vio-770 (Miltenyi Biotec, Germany).

Comment 8:

Methods: how was the activated PAR1 detected by flow cytometry? The antibody from Sigma is unconjugated. Was it stained with a secondary antibody or conjugated? Include more details.

Reply:

The percentage of FIIa-activated PAR1-positive (the antibody detects fragments of activated PAR1-cleaved-Ser42 protein) (Sigma-Aldrich, St. Louis, MO, USA) (visualized with goat anti-rabbit IgG Alexa 405; Thermo Fisher Scientific, Waltham, MA, USA) circulating PBMCs was measured using flow cytometry. We have added this information to the method section.

Comment 9:

Methods: Include the method for measurement of TNF and IL6 in plasma, this is hardly from medical records. Include also the methods for vWF:Ag, vWF:RCo and FVIII, same reason as for TNF and IL6 measurements.

Reply:

Routine laboratory results (accredited Labor Berlin - Charité Vivantes GmbH, Department of Laboratory Medicine) were obtained from the patient’s medical records, this include TNF-α (chemiluminescent immunoassay), IL-6 (electrochemiluminescence immunoassay), VWF:Ag, VWF:RCo (both latex-enhanced immunoassay), and FVIII (coagulometry). https://www.laborberlin.com/

Comment 10:

Methods: Explain how the thrombin activation was calculated. Is it really expressed as AU/L/minute (figure 1, 3, 5 and 6)? From the protocol supplied be the producer one would expect AU/L (or mL).

Reply:

Both you and reviewer #1 noted this concern. Therefore, we have adjusted the text to be clearer. FIIa activity in plasma was measured with the Human Thrombin Chromogenic Activity Kit (Assaypro LLC). The absorbance at 405 nm was read every 30 minutes for 2 hours. To generate a standard curve from the optimal reaction time, a graph was plotted using the standard concentrations on the x-axis and the change in absorbance per minute (∆A/min) on the y-axis after subtracting the background. The best fit line was determined by regression analysis of the 4-parameter curve. The unknown sample concentration was determined from the standard curve and the value was multiplied by the dilution factor. The conversion of WHO U and IU is 1 WHO U/ml = 1.2 IU/ml. The conversion of AU and IU is 1 AU/ml = 0.15 IU/ml.

Comment 11:

Figure 2: Units for CXCL4(PF4) and CXCL7 should be checked, probably it should be ng/mL not µg/mL. The presented plasma level of PF4 seems to be 15 times higher than what is expected in serum (and 1500 times higher than normal plasma).

Reply:

We have double-checked the results and came to the same conclusion as before. By checking numerous studies, we found a significant variety in reported results between the assays due to the lack of standardization.

Comment 12:

Figure 4: Both panel titles and figure legend need to be corrected. Quite confusing as presented. Left and right panel of C,  D and E each represent APCs, CD3- and lymphocytes, respectively. But at first glance it looks like this hold true for the left part of each panel. More other the terms “PAR1+” and “Activated PAR1” are both used in the figure but when reading the figure legend one gets the impression that both terms refer to PAR1 that has been cleaved. In the methods section a general PAR1-antibody is listed but has it been used? Is there actually any results or figure including flow cytometric results on PAR1-positiv cells (including cells positive for PAR1 independent of activation)? The figure legend needs some improvement – the “(A)” and “(B)” etc have not been placed correctly.

Reply:

We are grateful to know that our current approach requires some rethinking. In accordance with the reviewer’s important suggestion, we have added a more detailed description of flow cytometry (figure legend, method section) and have revised Figure 4 to be clearer.

Figure 4. Pro-inflammatory immune cell function in FDAF – Indications for PAR1 activation via thrombin. A: Increased plasma levels of pro-inflammatory mediators, namely tumor necrosis factor α (TNF-α) and interleukin 6 (IL-6), were seen in patients with FDAF. B: Phenotyping via flow cytometry of peripheral blood mononuclear cells (PBMCs) revealed a higher percentage (% of total PBMCs) of circulating pro-inflammatory cells that possess thrombin-activated (cleaved) PAR1 (TNF-α+ or IL-6+ cleaved PAR1+) in patients with FDAF. The antibody detects fragments of activated PAR1-cleaved-Ser42 protein. Cytokines TNF-α and IL-6 were visualized via intracellular staining of permeabilized cells. C–E: Evaluation of different immune cell subsets expressing PAR1 via flow cytometry. Antigen-presenting cells (APCs; HLA-DR+) (C), non-T cells (CD3-) (D), and lymphocytes (E). Left panel: percentage of a specified immune cell population expressing PAR1. Right panel: Increased mean fluorescence intensity (MFI) of PAR1. Extracellular (surface) PAR1 expression (= PAR1 that is accessible for activation via FIIa) was detected via staining of unpermeabilized PBMCs with a PAR1 antibody. Patients with FDAF (n = 80) were compared to controls (patients with chronic cardiovascular diseases but without AF) (n = 20). Results are expressed as single values, median, and p-values.

Comment 13:

Table 1: Include all events in the follow-up period, now only MACE has been listed. It should correspond to figure 6.

Reply:

We thank the reviewer for this assistant comment and accordingly added all MACEs.

Round 2

Reviewer 2 Report

Comments and Suggestions for Authors

I have no further comments.